# Fine-Tuning Large Language Models to Appropriately Abstain with Semantic Entropy

**Benedict Aaron Tjandra**[1*]   **Muhammed Razzak**[1*]   **Jannik Kossen**[1*]

**Kunal Handa**[1]   **Yarin Gal**[1]

[1] OATML, Department of Computer Science, University of Oxford

## Abstract

Large Language Models (LLMs) are known to hallucinate, whereby they generate plausible but inaccurate text. This phenomenon poses significant risks in critical applications, such as medicine or law, necessitating robust hallucination mitigation strategies. While recent works have proposed fine-tuning methods to teach LLMs to abstain from answering questions beyond their knowledge or capabilities, these methods rely on the existence of ground-truth labels or are limited to short-form responses. To address these limitations, we propose fine-tuning using semantic entropy, an uncertainty measure derived from introspection into the model which does not require external labels. We demonstrate that our approach matches or outperforms models fine-tuned using prior work and achieves strong performance for both short and long-form generations on a range of datasets.

## 1   Introduction

Large language models (LLMs) have made significant progress in natural language processing, achieving remarkable performance across a wide range of tasks (OpenAI, 2024; Meta, 2024; Google, 2024). Models, like GPT-4, Llama 3, and Gemini, have demonstrated capabilities that rival or surpass human performance in a range of domains and are increasingly deployed in the real world for various applications (Yang et al., 2023a). However, despite these advancements in performance, LLMs remain far from flawless, particularly when it comes to handling tasks that fall outside their scope of knowledge or reasoning abilities (Ji et al., 2023a; Huang et al., 2023), where they tend to exhibit *hallucinations*.

Hallucinations, in the context of LLMs, refer to instances where models generate content that, while appearing plausible, is factually inaccurate, contradicts previously established world knowledge, or does not make logical sense (Zhang et al., 2023). These hallucinations, while harmless in low-stakes applications, can have severe consequences when LLMs are deployed in safety-critical domains such as healthcare or legal services, where the generation of erroneous information could be costly or lead to real-world harm (Han et al., 2024; Weiser, 2023). As a result, mitigating hallucinations is crucial to ensuring the safety, trustworthiness, and overall reliability of LLMs (Rawte et al., 2023).

Various strategies have been proposed to mitigate hallucinations (Ji et al., 2023a; Tonmoy et al., 2024). Retrieval Augmented Generation (RAG) approaches attempt to ground LLM outputs by incorporating external knowledge sources (Lewis et al., 2021; Shuster et al., 2021; Ji et al., 2023b), anchoring responses in verified facts. Other techniques involve modifying the inference process to encourage more cautious generation (Shi et al., 2023; Chuang et al., 2024) or focus on post-hoc detection, where hallucinations are flagged after generation through methods such as confidence or uncertainty estimation (Kadavath et al., 2022; Azaria and Mitchell, 2023; Kuhn et al., 2023; Farquhar et al., 2024).

An approach that has gained attention due to its simplicity and effectiveness involves fine-tuning LLMs to abstain from answering questions that are outside the scope of their knowledge or

---

*Equal Contribution. Correspondence to aaron_tjandra@yahoo.com.

38th Conference on Neural Information Processing Systems (NeurIPS 2024), Safe Generative AI Workshop.

capabilities (Wen et al., 2024). To do this, an LLM is fine-tuned on a dataset that consists of examples of questions that the LLM should abstain from answering and, conversely, examples of questions that the LLM should willingly answer. Several recent works (Yang et al., 2023b; Cheng et al., 2024; Wolfe et al., 2024) teach the LLM to abstain from answering questions it answered incorrectly. These approaches, however, require access to ground-truth labels, which in many cases can be difficult or costly to obtain. Additionally, since ground-truth labels come from external sources, there is evidence that they are potentially noisy and hence not the most appropriate teaching signal Kossen et al. (2024). To avoid being dependent on ground-truth labels, Zhang et al. (2024) explored an uncertainty-based fine-tuning approach, R-Tuning-U, where they first approximate an LLM's uncertainty (as measured by entropy) in answering a question and subsequently training the LLM to abstain from answering questions it is uncertain. R-Tuning-U, however, is sensitive to the lexical and syntactical variations of generations, limiting its application to settings where the LLM is instructed to generate short-form responses consisting of not more than a few words, hindering its usefulness.

In this work, we propose to leverage semantic entropy (Kuhn et al., 2023; Farquhar et al., 2024) to overcome these limitations. Semantic entropy improves over R-Tuning-U by computing entropy over the semantic space that model generations occupy, rather than over raw token sequences. As semantic entropy evaluates the entropy over the semantic meaning of the generations, it is robust to lexical and syntactical variations of generations. This results in a better indicator of hallucinations in both short-form and long-form generation settings (Kuhn et al., 2023; Farquhar et al., 2024). By fine-tuning models to abstain from answering questions it is uncertain about using semantic entropy, we provide an approach to reduce the model's hallucinations, without relying on ground-truth labels or being restricted to the short-form response setting. This allows for a more flexible and reliable abstention across a wide range of tasks and domains, providing a powerful tool for reducing hallucinations.

We evaluate our method across several benchmarks and introduce a new metric called accuracy-engagement distance (AED), which quantifies model hallucinations more comprehensively by taking into account its *accuracy* and *engagement*, the number of questions it chooses to answer willingly. Using this metric, we show that models fine-tuned with semantic entropy significantly outperform R-Tuning and R-Tuning-U, existing approaches where the former is label-dependent and the latter is label-independent. Compared to R-Tuning and R-Tuning-U (Zhang et al., 2024), our method achieves up to 30.1% reduction in hallucination rates for long-form generations and up to 8.7% for short-form generations. Our method opens up new avenues to fine-tune models on both short-form and long-form generations without relying on ground-truth labels, making it easily scalable.

**Contributions.** In summary, the key contributions of this paper are:

- We demonstrate that models fine-tuned using semantic entropy (Section 5) match or outperform models fine-tuned using prior work, under both long-form (Long-QA) and short-form (Short-QA) answering settings (Section 6).
- We introduce the *accuracy-engagement distance (AED)*, a novel evaluation metric, that more holistically quantifies the extent of a model's hallucination by taking into account both the *accuracy* and *engagement* of a model (Section 4).

## 2   Related Work

There is a significant body of work in the literature that studies why hallucination occurs and how to prevent them. We redirect the reader to recent surveys (Ji et al., 2023a; Huang et al., 2023; Rawte et al., 2023; Chakraborty et al., 2024) for a comprehensive overview of hallucinations. For conciseness, we focus on previous work that are most pertinent to ours.

**Uncertainty Estimation in LLMs.** A variety of works have proposed uncertainty measures to detect hallucinations in LLM generations. Grey-box methods rely on token likelihoods and multiple samples to a prompt to measure LLM uncertainty (Kadavath et al., 2022; Kuhn et al., 2023; Farquhar et al., 2024; Zhang et al., 2024). White-box methods assume access to the internals of an LLM (weights and activations) and train models on the activations during generation to probe into the uncertainty of LLMs (Ahdritz et al., 2024; Kossen et al., 2024; Liu et al., 2024).

**Abstention Fine-Tuning.** There is a range of proposed methods to fine-tune LLMs to abstain from answering questions beyond their capabilities (Wen et al., 2024). Wolfe et al. (2024) looked into using QA datasets with unanswerable questions and fine-tuned LLMs to abstain from answering

those questions. Similarly, Brahman et al. (2024) developed a taxonomy for different abstention scenarios and, using prompting techniques, constructed a synthetic dataset to capture each abstention scenario. They then used this dataset to fine-tune models and measured their abstention rate in each abstention scenario. Yang et al. (2023b) and Cheng et al. (2024) sampled multiple responses to each question and leveraged the accuracy rate to fine-tune LLMs to abstain from questions whose accuracy rate is below a certain threshold. Zhang et al. (2024) proposed R-Tuning and R-Tuning-U to fine-tune LLMs under the Short-QA answering setting. In R-Tuning, models are instructed to generate single responses to a set of questions, and are fine-tuned to abstain from questions it answered incorrectly. In R-Tuning-U, multiple responses are sampled per question and the entropy of those responses is used to measure the uncertainty of an LLM to each question. The model is then fine-tuned to abstain on the top-50% most uncertain questions while keeping their original responses for the other half.

## 3 Background: Label-Free Uncertainty Estimation in LLMs

In this section, we familiarise the reader with R-Tuning-U and semantic entropy, two uncertainty quantifying methods for LLMs that we use to determine the set of questions an LLM should abstain from.

**R-Tuning-U.** R-Tuning-U (Zhang et al., 2024) uses an approximation of the *classical* conditional entropy given a particular question $q$ to measure an LLM's uncertainty. Let $G$ be the set of possible generations, where $\mathbf{s} \in G$ denotes a sequence of tokens with $s_i$ representing the $i$-th token in the sequence, and let $\mathbf{q}$ be the sequence of tokens obtained by tokenising our prompt $q$. Using the conditional probability distribution $p_\theta$ that we can obtain from an LLM, the probability of some token sequence $\mathbf{s}$ occurring given $\mathbf{q}$ is

$$p_\theta(\mathbf{s} \mid \mathbf{q}) = \prod_{i=1}^{|\mathbf{s}|} p_\theta(s_i \mid \mathbf{s}_{<i}, \mathbf{q}), \tag{1}$$

where $\mathbf{s}_{<i} = (s_1, ..., s_{i-1})$. The classical conditional entropy over generations is formally defined as

$$E(\mathbf{q}) = -\sum_{\mathbf{s} \in G} p_\theta(\mathbf{s} \mid \mathbf{q}) \log p_\theta(\mathbf{s} \mid \mathbf{q}). \tag{2}$$

However, since the size of $G$ may be extremely large, the above equation is intractable to compute exactly. R-Tuning-U arrives at a discrete approximation of Equation (2) by sampling $M$ generations and subsequently measuring the empirical probability of the occurrence of each generation. If we let $U$ to be the set of unique generations obtained from $S$, our list of sampled generations, and $c(u)$ to be the number of times $u \in U$ occurs in $S$, then R-Tuning-U approximates the classical entropy by

$$E(\mathbf{q}) \approx -\sum_{u \in U} \left( \frac{c(u)}{M} \right) \log \left( \frac{c(u)}{M} \right). \tag{3}$$

**Semantic Entropy.** The key drawback of R-Tuning-U is that it disregards the semantic meaning of the generations and is sensitive to lexical and syntactical variations. A model that generates "The capital of France is Paris" and "Paris is France's capital" when prompted twice in response to the question "What is France's capital?" is not uncertain in any meaningful sense (Kuhn et al., 2023). R-Tuning-U, however, will assign a non-zero uncertainty as the generations are different strings.

Semantic entropy improves over R-Tuning-U by taking into account the semantic meaning of generations. Instead of computing the entropy of the generations, it computes the entropy of *semantic equivalence classes* that the generations occupy, where a semantic equivalence class $C$ is defined to be the set of generations that share the same particular meaning. More concretely, letting $\mathcal{C}$ to be the set of all semantic equivalence classes, for any semantic equivalence class $C \in \mathcal{C}$, we have $\forall \mathbf{s}, \mathbf{s}' \in C \colon R(\mathbf{s}, \mathbf{s}')$, where $R(\cdot, \cdot)$ is a *semantic equivalence relation* that holds if and only if two generations have the same semantic meaning. Formally, the semantic entropy is defined as

$$SE(\mathbf{q}) = -\sum_{C \in \mathcal{C}} p_\theta(C \mid \mathbf{q}) \log p_\theta(C \mid \mathbf{q}), \tag{4}$$

where $p_\theta(C \mid \mathbf{q})$ is the probability of a semantic equivalence class $C$ occurring given $\mathbf{q}$:

$$p_\theta(C \mid \mathbf{q}) = \sum_{\mathbf{s} \in C} p_\theta(\mathbf{s} \mid \mathbf{q}) = \sum_{\mathbf{s} \in C} \prod_{i=1}^{|\mathbf{s}|} p_\theta(s_i \mid \mathbf{s}_{<i}, \mathbf{q}). \tag{5}$$

However, just as calculating classical entropy is intractable, calculating semantic entropy is equally intractable. Similar to R-Tuning-U, we can approximate the (discrete) semantic entropy by sampling $M$ generations in response to $q$ and using the number of generations in each semantic equivalence class (Farquhar et al., 2024) to approximate $p_\theta(C \mid \mathbf{q})$. Letting $C_1, \ldots, C_m$ to be the semantic equivalence classes that we can extract from our sampled list of generations $S$, the discrete approximation of semantic entropy is given by

$$SE(\mathbf{q}) \approx -\sum_{i=1}^{m} p_\theta(C_i \mid \mathbf{q}) \log p_\theta(C_i \mid \mathbf{q}) \approx -\sum_{i=1}^{m} \left( \frac{|C_i|}{M} \right) \log \left( \frac{|C_i|}{M} \right). \tag{6}$$

To compute the above equation in practice, we need to first operationalise the semantic equivalence relation $R(\cdot, \cdot)$. In this work, we follow Kuhn et al. (2023)'s approach to implement $R(\cdot, \cdot)$ using the concept of question-dependent bi-directional entailment. More specifically, we deem that two generations $\mathbf{s}$ and $\mathbf{s}'$ are semantically equivalent if an entailment model says they logically entail one another within the context of the question. We then use $R(\cdot, \cdot)$ to cluster $S$ into semantic equivalence classes, where each class consists of responses that share the same semantic meaning. Given these equivalence classes, we estimate the probability $p_\theta(C_i)$ of each class $C_i$ by dividing the number of responses in class $C_i$ by the total number of responses $M$. The semantic entropy for a question $q$ then follows via Equation (6). More details concerning the entailment model, semantic clustering, and semantic entropy calculation can be found in Appendix A.3.

## 4 Accuracy-Engagement Distance

Several works (Zhang et al., 2024; Feng et al., 2024) suggest using the accuracy over the set of willingly answered questions to measure the extent of hallucination of a model. Though a natural idea, we argue that this metric is not a holistic measure of model performance as it does not penalise the model from wrongly abstaining from a question. To illustrate, consider a model $\mathcal{A}_1$ that willingly answers all questions on a dataset of 2500 questions and attains an accuracy of 70%. Suppose that fine-tuning $\mathcal{A}_1$ yields a model $\mathcal{A}_2$ that willingly answers 10 questions and attains an accuracy of 70%. If we use accuracy to compare $\mathcal{A}_1$ and $\mathcal{A}_2$, then we would deem them as equivalent as they have the same accuracies. However, from a helpfulness point of view, this is misleading as $\mathcal{A}_2$ is clearly *worse* than $\mathcal{A}_1$ as it avoids answering a substantial number of questions that $\mathcal{A}_1$ previously got correct.

We can see that $\mathcal{A}_2$ has a **low engagement** as it abstains from answering a large number of questions. Ideally, our metric should **penalise low engagement and low accuracy** and reward **high engagement and high accuracy**. Adapting Tian et al. (2023)'s method to compare the truthfulness of biography generations, we propose to evaluate fine-tuned models using a novel evaluation metric, the *Accuracy-Engagement Distance*, that takes into account both the accuracy and engagement of a model.

Consider a fine-tuned model that answers $Q$ questions willingly. Among these $Q$ questions, the model answers $I$ questions incorrectly and $C$ questions correctly. We can conceptualise our fine-tuned model as occupying a single point in $\mathbb{R}^2$ whose coordinates are $(I, C)$. An ideal model, that has the highest accuracy and engagement, answers all questions correctly and is represented by the point $(0, |D|)$ in this space, where $|D|$ is the total number of questions in a particular dataset. The Accuracy-Engagement Distance (AED) is the normalised Euclidean distance between the point representing the fine-tuned model and the ideal model:

$$\text{AED} = \sqrt{\frac{I^2 + (|D| - C)^2}{2 \cdot |D|^2}}.$$

The AED ranges from 0 to 1 and is maximised when the model answers every question correctly (**max accuracy, max engagement**) and is minimised when the model answers every question incorrectly (**min accuracy, max engagement**). If we now compare $\mathcal{A}_1$ and $\mathcal{A}_2$ using AED, we can see that $\mathcal{A}_1$ achieves an AED of 0.30 while $\mathcal{A}_2$ achieves an AED of 0.71, penalising $\mathcal{A}_2$'s low engagement.

# 5 Abstention Fine-Tuning using Semantic Entropy

In this section, we introduce a fine-tuning strategy that leverages semantic entropy to enable model abstention in uncertain scenarios.

**Overview.** The key idea is to determine which questions to abstain from and to willingly answer based on the semantic entropy of a model's responses. Questions with high semantic entropy, indicating a high likelihood of a hallucination being generated, should be abstained from answering, while those with low semantic entropy should be answered with the model's standard response.

**Dataset Construction.** For each question in the training dataset, we generate a **standard response** using a low-temperature setting ($T = 0.1$) to encourage a deterministic output. Additionally, we generate $M = 10$ responses by sampling at a high temperature ($T = 1.0$) to capture the model's variability under more stochastic conditions.

The high-temperature responses are used to compute the semantic entropy as described in Section 3. Computing the semantic entropy of each question $q$ results in a set of $(q, SE(q))$ pairs where $SE(q)$ represents the semantic entropy of $q$'s responses. With the computed semantic entropy for each question, we partition the dataset into two subsets based on a user-defined uncertainty threshold $\tau$:

- **High-entropy set $H$:** This set contains questions where $SE(q) > \tau$, indicating a high level of uncertainty in the model's responses. For these questions, we modify the ground-truth label to an abstention phrase: *"I don't know the answer."*
- **Low-entropy set $L$:** This set includes questions with $SE(q) \leq \tau$, where the model is relatively confident. Here, the ground-truth label is set to be the model's standard response.

**Fine-Tuning Procedure.** Once the dataset is partitioned, we fine-tune the model using $H$ and $L$. We employ supervised fine-tuning with cross-entropy loss, where the model is trained to predict the next token in the concatenated input sequence (prompt + question + adjusted label). The model is encouraged to generate the standard response for questions in $L$, while abstaining for those in $H$.

Formally, given an answering setting prompt, i.e. Long-QA or Short-QA, tokenised questions $\mathbf{q}$, and their corresponding tokenised ground-truth labels $\mathbf{y}^{(q)}$ (either the standard response or abstention phrase), the model learns to minimise the following fine-tuning objective during training:

$$\mathcal{L}_{CE}(p_\theta) = - \sum_{q \in Q} \sum_{t=1}^{|\mathbf{y}^{(q)}|} \log p_\theta(y_t^{(q)} \mid \text{prompt}, \mathbf{q}, \mathbf{y}_{t-1}^{(q)}), \tag{7}$$

where $Q$ denotes the set of questions in the training set and $p_\theta(\cdot \mid \text{prompt}, \mathbf{q}, \mathbf{y}_{t-1}^{(q)})$ is the model's predicted next-token probability distribution given the answering setting prompt, question, and the first $t - 1$ tokens of the modified ground-truth label.

# 6 Experiments

We evaluate our abstention fine-tuning approach LLAMA-3-8B-INSTRUCT (Meta, 2024) across four datasets and two answering settings: 1) Long-QA, where we instruct the LLM to generate free-form sentence-length generations and 2) Short-QA, where we instruct the LLM to generate short-form answers. Our prompts for each answering setting can be found in Appendix A.1.

**Datasets.** We evaluate across four datasets: TriviaQA Joshi et al. (2017), BioASQ Tsatsaronis et al. (2015), NQ Kwiatkowski et al. (2019), and SQuAD Rajpurkar et al. (2016). We randomly select 2500 QA pairs from the validation split of each dataset and designate 2000 data points for training and 500 data points for in-distribution validation. We adopt a closed-book setting for our experiments and remove additional context that is present in TriviaQA and SQuAD. For each question, we use the model's standard response and the question's ground-truth label to assign an accuracy score to that question (Appendix A.2). As described in Section 5, the high-temperature responses are used to calculate the semantic entropy of each question. We compute semantic entropy with two entailment models resulting in two variants of semantic entropy: Semantic Entropy with DeBERTa entailment (SE (DeBERTa)) and Semantic Entropy with Llama-3-70B-Instruct entailment (SE (Llama)). Further implementation details can be found in Appendix A.3. In addition to semantic entropy, we use the high-temperature responses to compute the entropy for R-Tuning-U via a direct application of Equation (2).

**Fine-Tuning.** Our experiments involve fine-tuning a model on a training dataset and evaluating the fine-tuned model on the in-distribution validation split and out-of-distribution datasets that exclude the training set. For experiments concerning an uncertainty metric, i.e. R-Tuning-U, SE (DeBERTa), and SE (Llama), each fine-tuning run is associated with a threshold $\tau$, where we partition the training set into the high-entropy set and the low-entropy set. We then fine-tune the model via the method described in Section 5. For R-Tuning, we assign the set of incorrect questions to $H$ and the set of correct questions to $L$ and equivalently replace the ground-truth labels of $H$ with the abstention phrase and $L$ with the standard response of the model. Due to resource constraints, we use LoRA (Hu et al., 2021) to perform supervised fine-tuning. In addition to single dataset experiments, i.e. training on one dataset and evaluating on another dataset, we also conduct experiments where we train on multiple datasets – specifically by combining the training splits of TriviaQA, BioASQ, and NQ. We denote this setting as "Mult" in subsequent sections. In "Mult", the validation set is the combined validation sets of TrivaQA, BioASQ, and NQ, and the out-of-distribution set consists of SQuAD. Hyperparameter details can be found in Appendix A.4.

**Model Selection and Evaluation.** We conduct two forms of evaluation: 1) Best-Threshold Evaluation and 2) All-Threshold Evaluation. In **Best-Threshold Evaluation**, we conduct experiments with R-Tuning with 3 random seeds and report the mean and standard deviation of the AED on the in-distribution validation set and out-of-distribution datasets. For each of R-Tuning-U, SE (DeBERTa), and SE (Llama), we first train a different model on 9 equally-spaced thresholds, ranging from 0.25 and 2.25, and select the threshold $\tau$ that achieves the lowest AED on the in-distribution validation set. We conduct experiments with that threshold using 3 random seeds and report the mean and standard deviation AED on the in-distribution validation set and out-of-distribution datasets. This captures a realistic scenario where an in-distribution validation set is used to pick the threshold $\tau$ that leads to the lowest AED and subsequently evaluating how well this fine-tuned model generalises on out-of-distribution settings.

In **All-Threshold Evaluation**, we evaluate each fine-tuned model trained at each of the 9 thresholds in our *single-dataset experiments*, recording the number of incorrect and correct questions each fine-tuned model gets on out-of-distribution datasets. To inspect the formation of any overall trends, we aggregate the number of incorrect and correct questions to build an "adaptation" plot, where each point represents the average incorrect and correct responses a fine-tuned model gets on an out-of-distribution dataset when trained at a specific threshold. For both types of evaluation, we perform greedy inference ($T = 0$).

# 7    Results and Discussion

This section presents and discusses our results for Best-Threshold and All-Threshold Evaluation.

**Models Fine-Tuned on Semantic Entropy Have Lower AEDs.** Figure 1 and Figure 2 presents results for Best-Threshold Evaluation. Figure 1 shows in-distribution experiments where we average results across different seeds, while Figure 2 shows out-of-distribution experiments where we average results across different seeds *and* further averaging experiments that share the same out-of-distribution dataset. A granular breakdown of our out-of-distribution experiments can be found in Appendix A.6. We further aggregate the means of all datasets and present the overall average for each setting in Table 1. From Figure 1 and Figure 2, fine-tuning on semantic entropy, under both Long-QA and Short-QA answering settings and on in-distribution and out-of-distribution evaluations, yields models with AEDs that are typically equal or lower than models fine-tuned with R-Tuning and R-Tuning-U. We also see that fine-tuning on semantic entropy computed with a stronger entailment model (SE (Llama)) largely led to models with lower AEDs. Moreover, from

Table 1: SE (Llama) and SE (DeBERTa) achieves the lowest overall average Accuracy-Engagement Distances for both Long-QA and Short-QA. **Bold** indicates lowest.

| Setting | | SE (Llama) (ours) | SE (DeBERTa) (ours) | R-Tuning | R-Tuning-U | Original Model |
|---|---|---|---|---|---|---|
| Long-QA: | In-Distribution | **0.364** | 0.411 | 0.399 | 0.521 | 0.380 |
| Short-QA: | In-Distribution | 0.428 | **0.427** | 0.469 | 0.438 | 0.467 |
| Long-QA: | Out-of-Distribution | **0.406** | 0.432 | 0.438 | 0.541 | 0.425 |
| Short-QA: | Out-of-Distribution | **0.473** | 0.482 | 0.508 | 0.485 | 0.525 |

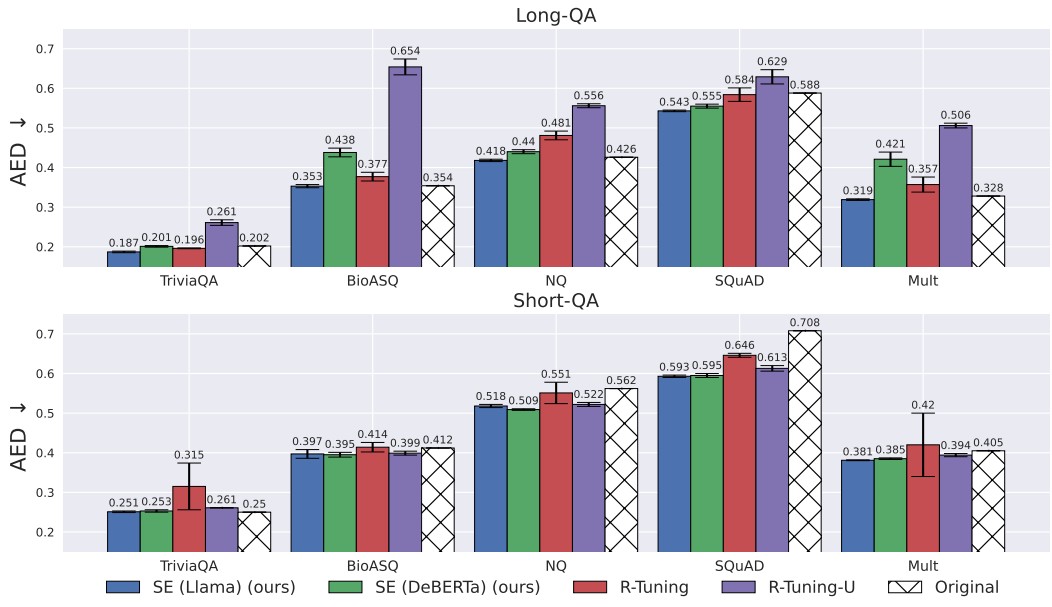

Figure 1: Our method, SE (Llama), matches or outperforms R-Tuning and R-Tuning-U for Long-QA and Short-QA in in-distribution experiments. Mean Accuracy-Engagement Distances (AEDs) are shown on top of each bar. Standard deviations are shown as error bars. The lower the AED, the better.

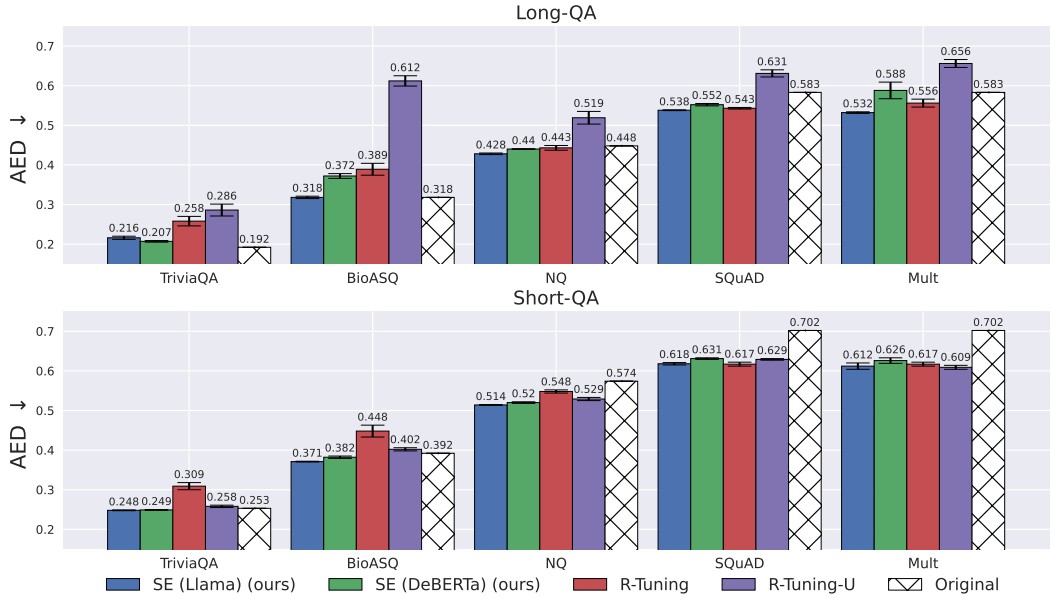

Figure 2: Our method, SE (Llama), matches or outperforms R-Tuning and R-Tuning-U for Long-QA and Short-QA in out-of-distribution experiments. Mean Accuracy-Engagement Distances (AEDs) are shown on top of each bar. Standard deviations are shown as error bars. The lower the AED, the better.

Table 1, we observe that, in aggregate, fine-tuning on R-Tuning and R-Tuning-U leads to models that often are *worse* than the original model. In contrast, fine-tuning on SE (Llama) yields models that **significantly outperform** existing methods and the original model. Notably, if we compare SE (Llama) with R-Tuning and R-Tuning-U, we obtain up to 30.1% and 8.7% reduction in AEDs for in-distribution experiments for Long-QA and Short-QA respectively. For out-of-distribution experiments, we obtain reductions up to 25.0% and 6.9% for Long-QA and Short-QA.

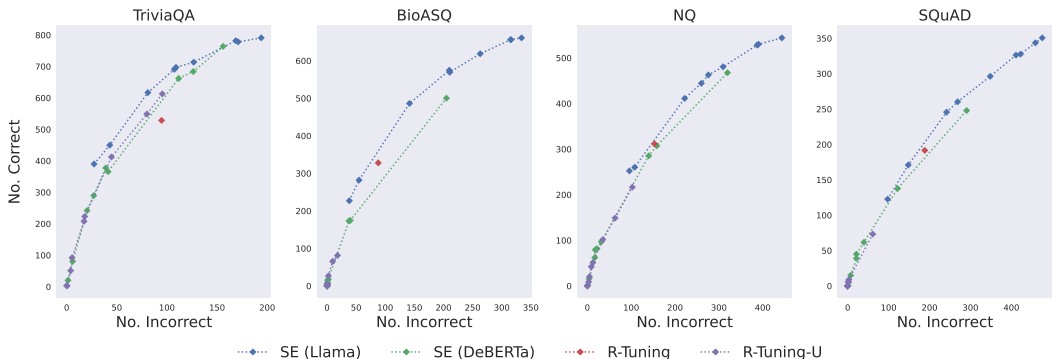

Figure 3: SE (Llama) forms a frontier over other methods in the Long-QA Adaptation Plot. Each point represents a fine-tuned model trained at a specific threshold.

**Fine-Tuning with Semantic Entropy Is a Pareto Improvement For Long-QA.** We present the Long-QA adaptation plot in Figure 3 as a result of All-Threshold Evaluation. Here, we observe that **models fine-tuned with SE (Llama) form a frontier over models fine-tuned using other methods**. Since the AED is the Euclidean distance from an ideal model that has maximum accuracy and engagement (the top left corner of the adaptation plot), then it follows that **models fine-tuned with SE (Llama) attains lower AEDs no matter the uncertainty threshold on out-of-distribution settings for Long-QA**. This further underscores the effectiveness of semantic entropy as an abstention fine-tuning method. It is difficult, however, to discern an overall trend in the Short-QA adaptation plot, which we show at Appendix A.5, where it seems that no method forms a frontier above another. This may be because there are lesser lexical and syntactical variances in Short-QA than in Long-QA, which leads to weaker methods such as SE (DeBERTa) and R-Tuning Entropy performing equivalently as SE (Llama) at numerous thresholds.

**Takeaways.** We can see models fine-tuned with semantic entropy, in most cases, outperform models fine-tuned with R-Tuning and R-Tuning-U, which indicates that semantic entropy is a clearer signal to reduce hallucinations than a model's correctness on a question and classical entropy. This is because semantic entropy is a more delineative statistic of a model's uncertainty than both R-Tuning and R-Tuning-U, which facilitates model learning and generalisation. Our findings represent an advancement in fine-tuning methodologies for both Long-QA and Short-QA answering settings, opening new avenues for reducing hallucinations for both short-form and long-form generations without the reliance on exhaustively labeled datasets.

**Limitations.** We note that in some instances of our single dataset experiments, the original model still attains lower AEDs than the fine-tuning approaches we have explored in our experiments. Due to resource constraints, we suspect that this is due to the relatively low LoRA rank $r = 8$ employed during training and the relatively small number of data points (2000) that we have used to train our models. Indeed, despite convergence on the training set, we observe that our fine-tuned models cannot exactly fit the training set. Future work could include full fine-tuning and to scale up our experiments to include more training points. We would also like to improve the reliability of our adaptation plots by repeating each threshold experiment multiple times.

## 8    Conclusion

In this work, we proposed using semantic entropy to fine-tune LLMs to abstain from answering questions beyond their capabilities. Under our proposed evaluation metric, the Accuracy-Engagement Distance which accounts for both the accuracy and engagement of a model, we demonstrated that models fine-tuned on semantic entropy matched or outperformed models fine-tuned via existing methods that relied on ground-truth labels or classical entropy.

Other than scaling up our experiments as discussed previously, future work may entail experimenting with other open-source LLMs to see if our conclusions have a generalising effect. Furthermore, future work can adapt our work to apply to longer generations, i.e. paragraphs or biographies. Finally, given the success of using semantic entropy to reduce hallucinations and recent evidence

that the model may be computing semantic entropy internally (Kossen et al., 2024), future work can also explore using semantic entropy as a fine-tuning method to calibrate models and comparing its viability with previous works (Zhang et al., 2024; Kadavath et al., 2022).

**Acknowledgments.** The authors thank Neil Band and members of the OATML group for insightful discussions throughout the development of this paper.

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

# A   Appendix / supplemental material

## A.1   Answering Setting Prompts

### A.1.1   Long-QA Answering Setting

In the Long-QA setting, we instruct models to generate free-form, sentence-length responses using the following prompt:

> Answer the following question in a single complete sentence. Short-form answers without a proper subject and verb are not allowed. There should be a subject, verb, and an object in your complete sentence and your answers should address the question directly.
> Question: {{ Insert Question }}
> Answer:

Figure 4: Long-QA Free-form Prompt.

### A.1.2   Short-QA Answering Setting

In the Short-QA setting, we use the following prompt to instruct the model to generate short-form responses not more than a few words:

> Answer the following question as briefly as possible.
> Question: Which chemical element has the chemical symbol Ca?
> Answer: Calcium
>
> Question: How many TAp73 isoforms have been identified in humans?
> Answer: Seven
>
> Question: What is the powerhouse of the cell?
> Answer: Mitochondria
>
> Question: Who authored the Harry Potter book series?
> Answer: J.K. Rowling
>
> Question: What countries is the G7 made up of?
> Answer: Canada, France, Germany, Italy, Japan, the United Kingdom and the United States.
>
> Question: {{ Insert Question }}
> Answer:

Figure 5: Prompt for Short-QA.

## A.2 Accuracy Evaluation

Given a standard response of our model to a question $q$, and the ground-truth label for that question, we use LLAMA-3-8B-INSTRUCT and the following prompt to assess the accuracy of the standard response:

> We are assessing the quality of answers to the following question: {{ Insert Question }}
> The following are expected answers to this question: {{ Insert Ground-Truth Labels }}
> The proposed answer is: {{ Insert Generation }}
> Within the context of the question, does the proposed answer mean the same as any of the expected answers?
> Respond only with yes or no.
> Response:

Figure 6: Prompt template for accuracy evaluation.

## A.3 Semantic Entropy Implementation

The non-trivial part of computing the semantic entropy lies in computing the semantic equivalence relation $R(\cdot, \cdot)$ that is true if two generations share the same semantic meaning (Section 3). While there are potentially many choices to implement $R(\cdot, \cdot)$, in this work, we follow Kuhn et al. (2023) in using the idea of bi-directional entailment to determine semantic equivalence. Here, we treat two generations $s$ and $s'$ to be semantically equivalent if and only if $s$ logically entails $s'$ and vice versa. For example, 'The capital of France is Paris' and 'Paris is the capital of France' share the same meaning as they logically entail each other. However, as we are supplied with a question, we must constrain our entailment to hold within the context of the question. For example, the generations 'Paris' and 'The capital of France is Paris' on their own do not entail one another as the former only declares 'Paris', without stating that it is the capital of France. However, if the generations were produced with respect to the question 'What is the capital of France?', then within the context of the question, both generations entail one another.

As described in Section 6, we assign two variants of semantic entropy to each question, SE (DeBERTa) and SE (Llama), each using different language models to perform entailment. The first of these uses DeBERTa-NLI, which follows Kuhn et al. (2023)'s proposal, and the second of these uses Llama-3-70B-Instruct, which is inspired from Farquhar et al. (2024)'s findings that LLMs can perform entailment well.

### A.3.1 Entailment using DeBERTa-NLI

DeBERTa-NLI (He et al., 2021) is a language model based on the transformer encoder-decoder architecture that is fine-tuned on the task of natural language inference (NLI). In NLI, we are given a 'premise' and a 'hypothesis', and the task is to classify whether the hypothesis logically follows from the premise (entailment), logically contradicts the premise (contradiction), or is logically undetermined given the premise (neutral). To use DeBERTa-NLI to see whether two generations $s$ and $s'$ entail one another given the question, we first concatenate the question to each $s$ and $s'$ and then concatenate both concatenations together using a special token. DeBERTa-NLI then classifies this concatenation as 'entailment', 'contradiction', or 'neutral'. We next do this for the other direction and only deem that $s$ and $s'$ are semantically equivalent if DeBERTa-NLI says 'entailment' for both directions.

### A.3.2 Entailment using Llama-3-70B-Instruct

Llama-3-70B-Instruct is the 70 Billion-parameter variant of Llama-3-8B-Instruct. Leveraging Llama-3-70B-Instruct's ability to follow instructions and to perform NLP tasks through in-context learning, we used a 5-shot ICL prompt to do question-dependent (uni-directional) NLI (Appendix A.3.2). Equivalently to the above, we only deem two generations $s$ and $s'$ as semantically equivalent if Llama-3-70B-Instruct produces 'entailment' for both directions.

We are given two possible answers to a question, "Possible Answer 1" and "Possible Answer 2". In this task, we are trying to evaluate whether "Possible Answer 1" semantically entail "Possible Answer 2".

Question: "By what name is singer 'Anthony Dominic Benevetto' better known?"
Possible Answer 1: The singer Anthony Dominic Benevetto is better known as Toni Basil.
Possible Answer 2: The singer Anthony Dominic Benevetto is better known as Antonio Carlos Jobim.
Does Possible Answer 1 semantically entail Possible Answer 2? Respond with only one word: entailment, contradiction, or neutral.
Answer: contradiction

Question: "Which wife of Henry VIII had already married twice before she became queen, and married for a fourth time after Henry's death?"
Possible Answer 1: Anne Boleyn is the wife of Henry VIII.
Possible Answer 2: Anne Boleyn is the answer.
Does Possible Answer 1 semantically entail Possible Answer 2? Respond with only one word: entailment, contradiction, or neutral.
Answer: entailment

Question: "Who did Simple Simon meet on his way to the fair?"
Possible Answer 1: He met a pie-man.
Possible Answer 2: He met the following: a pie-man, a horse, a cow, and a fox.
Does Possible Answer 1 semantically entail Possible Answer 2? Respond with only one word: entailment, contradiction, or neutral.
Answer: neutral

Question: "The most northerly part of mainland Australia is in which state?"
Possible Answer 1: Queensland is the most northerly part of mainland Australia.
Possible Answer 2: The most northerly part of mainland Australia is Western Australia.
Does Possible Answer 1 semantically entail Possible Answer 2? Respond with only one word: entailment, contradiction, or neutral.
Answer: contradiction

Question: "The most northerly part of mainland Australia is in which state?"
Possible Answer 1: It is in Queensland, in Western Australia.
Possible Answer 2: Queensland.
Does Possible Answer 1 semantically entail Possible Answer 2? Respond with only one word: entailment, contradiction, or neutral.
Answer: entailment

Question: "David Jason starred as Inspector Frost, but who played his boss Superintendent Norman Mullet?"
Possible Answer 1: Stephen McGann played his boss.
Possible Answer 2: Norman Mullet played his boss in Superintendent Norman Mullet.
Does Possible Answer 1 semantically entail Possible Answer 2? Respond with only one word: entailment, contradiction, or neutral.
Answer: contradiction

Question: {{ Insert Question }}
Possible Answer 1: {{ Insert s }}
Possible Answer 2: {{ Insert s' }}
Does Possible Answer 1 semantically entail Possible Answer 2? Respond with only one word: entailment, contradiction, or neutral.
Answer:

Figure 7: Uni-directional entailment ICL prompt.

Having shown how to implement the semantic equivalence relation $R$ via one of the two methods above, we can now cluster the high-temperature responses into semantic equivalence classes $C_1, \ldots, C_m$. The discrete semantic entropy then follows via direct calculation of Equation (6). We show a concrete implementation of semantic clustering and the subsequent calculation of discrete semantic entropy in Code Block 1.

```python
from collections import Counter
def semantic_entropy(question, high_temp_generations):
    next_id = 0
    assignment = [-1] * len(high_temp_generations)
    for i, s1 in enumerate(high_temp_generations):
        if assignment != -1:
            continue
        # If s1 has not been assigned an id, assign it next_id.
        assignment[i] = next_id
        for j, s2 in enumerate(high_temp_generations[i + 1:]):
        if is_semantically_equivalent(question, s1, s2):
                assignment[j] = assignment[i]
        next_id += 1
```

```
14    freq_list = list(Counter(assignment.values()))
15    semantic_entropy = scipy.stats.entropy(freq_list)
16    return semantic_entropy
```

Code Block 1: Python code to compute discrete semantic entropy.

## A.4 Hyperparameter Details

All of our experiments under both answering settings use a global hyperparameter configuration. We found that a learning rate of $3 \times 10^{-5}$, a batch size of 48, and training for 7 epochs under a cosine annealing schedule with a cycle of 0.2 yields decent convergence on the training set and the in-distribution validation set. We employed the AdamW optimiser (Loshchilov and Hutter, 2019) for all experiments and used LoRA with $r = 8$ on the query and value projection matrices.

## A.5 Short-QA Out-of-Distribution Adaptation Plot

We present the Short-QA adaptation plot in Figure 8. As we can see, the effect is less pronounced than that of the Long-QA adaptation plot and it is harder to discern an overall trend.

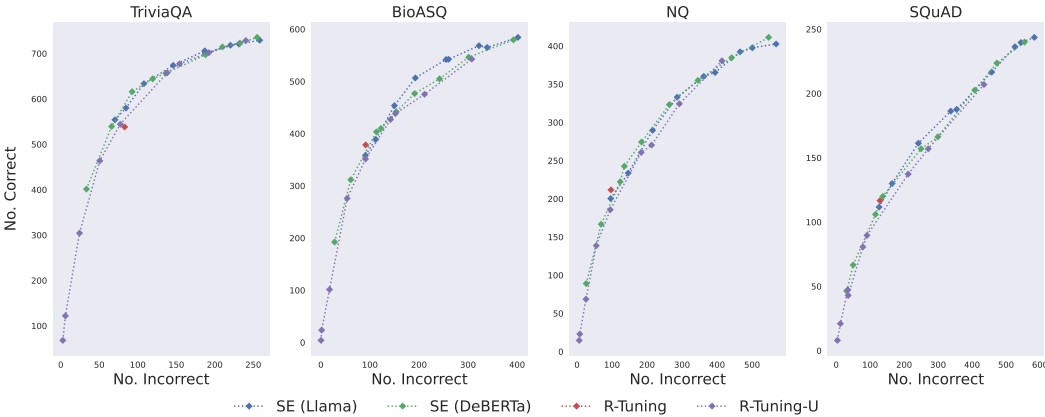

Figure 8: Short-QA Out-of-Distribution Adaptation Plot. Each point represents a fine-tuned model trained at a specific threshold.

## A.6 Granular Breakdown of Experiments

Table 2 shows a granular breakdown of our experiments for Best-Threshold Evaluation. Here, the **Setting** column denotes how the model is trained and evaluated. For example, "Long-QA: TriviaQA ∘ TriviaQA" means that the model is trained under Long-QA, on the training split of TriviaQA, and evaluated on the in-distribution validation split of TriviaQA. "Short-QA: BioASQ ∘ SQuAD" means that the model is trained on the training split of BioASQ and evaluated on all SQuAD data points under the Short-QA answering setting.

Table 2: Granular Breakdown of Experiments: Long-QA and Short-QA mean $\pm$ standard deviation Accuracy-Engagement Distances for each fine-tuning method, the lower the better. Green indicates lowest, blue indicates second-lowest.

| Setting | | | R-Tuning | R-Tuning-U | SE (DeBERTa) (ours) | SE (Llama) (ours) | Original Model |
|---|---|---|---|---|---|---|---|
| Long-QA: | TriviaQA | ∘ TriviaQA | 0.196±0.001 | 0.261±0.007 | 0.201±0.002 | 0.187±0.002 | 0.202 |
| Long-QA: | TriviaQA | ∘ BioASQ | 0.313±0.002 | 0.562±0.018 | 0.322±0.002 | 0.310±0.002 | 0.318 |
| Long-QA: | TriviaQA | ∘ NQ | 0.424±0.001 | 0.491±0.008 | 0.443±0.001 | 0.432±0.003 | 0.448 |
| Long-QA: | TriviaQA | ∘ SQuAD | 0.532±0.004 | 0.615±0.009 | 0.557±0.008 | 0.541±0.001 | 0.583 |
| Long-QA: | BioASQ | ∘ TriviaQA | 0.213±0.001 | 0.268±0.032 | 0.203±0.003 | 0.210±0.002 | 0.192 |
| Long-QA: | BioASQ | ∘ BioASQ | 0.377±0.011 | 0.654±0.020 | 0.438±0.011 | 0.353±0.004 | 0.354 |
| Long-QA: | BioASQ | ∘ NQ | 0.427±0.002 | 0.541±0.037 | 0.432±0.001 | 0.424±0.002 | 0.448 |
| Long-QA: | BioASQ | ∘ SQuAD | 0.538±0.000 | 0.628±0.025 | 0.544±0.001 | 0.531±0.001 | 0.583 |
| Long-QA: | NQ | ∘ TriviaQA | 0.267±0.014 | 0.339±0.020 | 0.212±0.002 | 0.216±0.003 | 0.192 |
| Long-QA: | NQ | ∘ BioASQ | 0.403±0.020 | 0.658±0.008 | 0.404±0.012 | 0.318±0.002 | 0.318 |
| Long-QA: | NQ | ∘ NQ | 0.481±0.011 | 0.556±0.005 | 0.440±0.005 | 0.418±0.003 | 0.426 |
| Long-QA: | NQ | ∘ SQuAD | 0.560±0.006 | 0.651±0.006 | 0.554±0.002 | 0.541±0.001 | 0.583 |
| Long-QA: | SQuAD | ∘ TriviaQA | 0.293±0.034 | 0.252±0.026 | 0.206±0.004 | 0.223±0.012 | 0.192 |
| Long-QA: | SQuAD | ∘ BioASQ | 0.451±0.041 | 0.617±0.033 | 0.389±0.012 | 0.327±0.007 | 0.318 |
| Long-QA: | SQuAD | ∘ NQ | 0.478±0.019 | 0.526±0.030 | 0.444±0.004 | 0.427±0.003 | 0.448 |
| Long-QA: | SQuAD | ∘ SQuAD | 0.584±0.017 | 0.629±0.018 | 0.555±0.005 | 0.543±0.002 | 0.588 |
| Long-QA: | Triv,Bio,Nq | ∘ Triv,Bio,Nq | 0.357±0.019 | 0.506±0.006 | 0.421±0.018 | 0.319±0.002 | 0.328 |
| Long-QA: | Triv,Bio,Nq | ∘ SQuAD | 0.556±0.010 | 0.656±0.010 | 0.588±0.021 | 0.532±0.002 | 0.583 |
| Short-QA: | TriviaQA | ∘ TriviaQA | 0.315±0.059 | 0.261±0.000 | 0.253±0.003 | 0.251±0.002 | 0.250 |
| Short-QA: | TriviaQA | ∘ BioASQ | 0.398±0.007 | 0.382±0.005 | 0.382±0.006 | 0.369±0.002 | 0.392 |
| Short-QA: | TriviaQA | ∘ NQ | 0.545±0.010 | 0.520±0.003 | 0.524±0.003 | 0.515±0.000 | 0.574 |
| Short-QA: | TriviaQA | ∘ SQuAD | 0.612±0.003 | 0.612±0.002 | 0.604±0.003 | 0.608±0.002 | 0.702 |
| Short-QA: | BioASQ | ∘ TriviaQA | 0.273±0.001 | 0.252±0.002 | 0.254±0.002 | 0.246±0.001 | 0.253 |
| Short-QA: | BioASQ | ∘ BioASQ | 0.414±0.012 | 0.399±0.005 | 0.395±0.006 | 0.397±0.011 | 0.412 |
| Short-QA: | BioASQ | ∘ NQ | 0.528±0.003 | 0.521±0.003 | 0.521±0.002 | 0.511±0.004 | 0.574 |
| Short-QA: | BioASQ | ∘ SQuAD | 0.611±0.003 | 0.657±0.006 | 0.654±0.005 | 0.618±0.007 | 0.702 |
| Short-QA: | NQ | ∘ TriviaQA | 0.327±0.026 | 0.250±0.002 | 0.243±0.001 | 0.246±0.001 | 0.253 |
| Short-QA: | NQ | ∘ BioASQ | 0.479±0.042 | 0.402±0.009 | 0.379±0.002 | 0.373±0.003 | 0.392 |
| Short-QA: | NQ | ∘ NQ | 0.551±0.027 | 0.522±0.005 | 0.509±0.002 | 0.518±0.004 | 0.562 |
| Short-QA: | NQ | ∘ SQuAD | 0.629±0.013 | 0.619±0.001 | 0.634±0.003 | 0.628±0.003 | 0.702 |
| Short-QA: | SQuAD | ∘ TriviaQA | 0.326±0.006 | 0.271±0.009 | 0.249±0.001 | 0.252±0.002 | 0.253 |
| Short-QA: | SQuAD | ∘ BioASQ | 0.468±0.010 | 0.422±0.008 | 0.384±0.005 | 0.372±0.001 | 0.392 |
| Short-QA: | SQuAD | ∘ NQ | 0.570±0.003 | 0.546±0.011 | 0.514±0.004 | 0.516±0.002 | 0.574 |
| Short-QA: | SQuAD | ∘ SQuAD | 0.646±0.005 | 0.613±0.007 | 0.595±0.005 | 0.593±0.003 | 0.708 |
| Short-QA: | Triv,Bio,Nq | ∘ Triv,Bio,Nq | 0.420±0.080 | 0.394±0.004 | 0.385±0.002 | 0.381±0.001 | 0.405 |
| Short-QA: | Triv,Bio,Nq | ∘ SQuAD | 0.617±0.005 | 0.609±0.005 | 0.626±0.007 | 0.612±0.008 | 0.702 |

