# OpenReview forum: "Fine-Tuning Large Language Models to Appropriately Abstain with Semantic Entropy"
_NeurIPS.cc/2024/Workshop/SafeGenAi — SafeGenAi Poster_

### Official Review · Reviewer_Yrxp · 2024-10-09
**The paper combines ideas from certainty calibration, hallucination detection, and fine-tuning (like LoRA) to address LLM hallucination uncertainty, an important safety issue. A key strength is its integration of various techniques, but its weakness is limited testing, only applying the fine-tuning method to QA, which is just one area where hallucinations occur.**

**Rating:** 6
**Confidence:** 4

**Review:**

## Summary

This paper introduces a novel approach to hallucination mitigation in large language models through an unsupervised fine-tuning technique that does not require labeled data. The method defines hallucinations as arising from the model's uncertainty regarding learned facts, using semantic entropy to assess this uncertainty across various domains. A key innovation is the introduction of the **Accuracy Engagement Distance (AED)**, a new evaluation metric that considers the model's willingness or confidence in generating correct answers, particularly relevant in scenarios with longer contexts. The approach is tested in long-context QA settings, where the fine-tuned model demonstrates significant improvements in both hallucination reduction and answer accuracy.

## Strengths

- Tackles a critical challenge in the safety tuning of large language models (LLMs), specifically focusing on hallucination mitigation.
- Provides a comprehensive introduction to semantic entropy, highlighting its importance and the limitations of current approaches in reducing hallucinations.
- Introduces the novel concept of **Accuracy-Engagement Distance (AED)**, which offers valuable insights into assessing LLM performance and has potential for broader application in future research.
- Emphasizes the necessity of in-distribution and out-of-distribution testing for fine-tuning and training techniques across diverse domains, thoroughly addressing this aspect in the proposed work.

## Weaknesses

- AED is not compared to other SOTA hallucination detection metrics such as EigenScore and is said to have a more “holistic” capability of quantifying hallucination but there is no proof or experiment to support that. EigenScore can be found in the following paper:

    https://arxiv.org/abs/2402.03744

- Willingness is an important concept to be discussed when it comes to hallucinations and the paper well discusses it. However, an important question is remained undiscussed: What does actually willingness mean? Is willingness supposed to be coded by humans within the test set?
- Seems like the fine-tuning protocol is doing better on Long-QA which sort of shows the model does better in general if it is provided a longer context, however, if the context is not that long, i.e., short-QA, there is not much difference and normal fine tuning is doing better. This suggests that for simpler tasks, this fine-tuning process may not provide significant benefits and could lead to inefficient use of resources.

## Questions

- “We evaluate our method across several benchmarks and introduce a new metric called Accuracy Engagement distance (AED), which quantifies model hallucinations more comprehensively by taking into account its accuracy and engagement, the number of questions it chooses to answer willingly.” what does the last part of the sentence mean?
- what does accuracy mean in Section 4? is it the quantity of a hallucination detection metric? Or actual accuracy of the model on a test set? As far as the literature describes, hallucination which is the aim of the paper to minimize, is different from accuracy on a test set. Even though the argument in section 4 about not favouring accuracy over willingness to answer makes sense in the context of QA, it might not on other definitions of hallucination.
- What is the best way of determining $\tau$ for the experiments?
- Do you think using LLMs to evaluate whether other LLMs are hallucinating (in this case for the semantic equivalence class analysis) is a good idea?

## Limitations

- The use of semantic entropy relies on an external entailment model to create equivalence classes, which introduces additional inference latency and may reduce overall efficiency.
- The Accuracy-Engagement Distance (AED) metric requires either human annotation or an additional model to evaluate whether the LLM is abstaining from answering, adding complexity and inefficiency to the fine-tuning process.
- While the fine-tuning protocol performs well on long-context QA tasks, indicating that the model benefits from extended context, it shows little improvement on short-QA tasks, where standard fine-tuning methods outperform it. Additionally, there is an ablation study on different values of τ\tauτ, but no guidance on how to select the optimal value for abstention fine-tuning in practice.

## Soundness

Very sound and there are well structured arguments except when the holistic-ness of AED is discussed. The mathematical notations well resonate with what the paper is presenting.

## Presentation

Good presentation. No major flaws.

## Contribution

Novel contribution of using the internals of a large language model and semantic entropy in fine tuning for hallucination mitigation.

## Overall

In general the paper brings together multiple ideas used in certainty calibration, hallucination detection, and fine tuning (such as LoRA) to solve a problem of uncertainty in large language model hallucinations which is a high priority concern within safety.
The method could be experimented against other post hoc methods to provide an evidence of extensive usability and the new proposed metric could be compared against other hallucination detection metric. The only setting in which this fine tuning is applied is QA which is only a small subset of where hallucinations occur in LLMs.

---

### Official Review · Reviewer_Qmnz · 2024-10-09
**Fine-Tuning Large Language Models to Appropriately Abstain with Semantic Entropy**

**Rating:** 7
**Confidence:** 4

**Review:**

Summary:

This paper provides a new method to fine-tune models. They propose having the LLM to avoid answering questions it is uncertain about using semantic entropy, in hopes of reducing the model’s hallucinations. They propose that questions with high semantic entropy, which show a higher likelihood of hallucinations, should not be answered by the model, while questions with low semantic entropy can be.

Additionally, they propose a new metric named Accuracy Engagement Distance (AED), which quantifies model hallucinations by taking into account its accuracy and engagement, or the number of questions the model willingly answers. Using the AED, they illustrate that models fine-tuned on semantic entropy either matched or outperformed models fine-tuned using previous methods, which relied on ground-truth labels.

Experiments are conducted across four datasets (TriviaQA, BioASQ , NQ, and SQuAD) with both long and short-form generations. Their method achieves up to 30.1% reduction in hallucination rates for long-form generations and up to 8.7% for short-form generations in comparison to other methods such as R-Tuning and R-Tuning-U.

Pros:
- Method is able to reduce hallucinations for generations without the reliance on labeled datasets.
- Accuracy Engagement Distance (AED) is a better metric than previous ones as it takes into account helpfulness, which penalizes low engagement.
- Very thorough experiments

Cons:
- Results in Short-QA look a little underwhelming

---

### Official Review · Reviewer_CY3Q · 2024-10-09
**Well written paper, nice work**

**Rating:** 8
**Confidence:** 4

**Review:**

Pros
1. This paper utilizes semantic entropy to compute a novel and practical metric that helps mitigating hallucinations.
2. The paper is well written and the experiments are all nicely visualized.

Cons
1. The authors did mention some results can be further improved. Increasing dataset size by 5 times (i.e. 2k to 10k for training) may lead to results that can further strengthen this work.